# Ionic Liquids Enhanced Alkynyl Schiff Bases Derivatives of Fipronil Synthesis and Their Cytotoxicity Studies

**DOI:** 10.3390/molecules24183223

**Published:** 2019-09-04

**Authors:** Xiu Liu, Linya Huang, Hongjun Chen, Na Li, Chao Yan, Chenzhong Jin, Hanhong Xu

**Affiliations:** 1Key Laboratory of Pesticide Harmless Application, Collaborative Innovation Center for Field Weeds Control (CICFWC) of Hunan Province, Hunan University of Humanities, Science and Technology, Loudi 417000, China; 2Key Laboratory of Natural Pesticide and Chemical Biology, Ministry of Education, South China Agricultural University, Guangzhou 510642, China; 3Hunan Provincial Key Laboratory of Fine Ceramics and Powder Materials, School of Materials and Environmental Engineering, Hunan University of Humanities, Science and Technology, Loudi 417000, China

**Keywords:** *N*-arylpyrazole derivatives, schiff base, cytotoxicity, fipronil derivatives

## Abstract

To obtain highly selective toxic derivatives of fipronil, a series of Schiff bases with an alkynyl group (**3a**–**3k**) were designed and synthesized from 4-ethynylbenzaldehyde (**2**) and 4-substituted 5-amino-*N-*arylpyrazole (**1a**–**1k**) via a nucleophilic addition elimination reaction in ionic liquids. Utilization of ionic liquids was demonstrated to endow the yield of each compound beyond 50%, which was enhanced over 1.5 times of the synthetic productive rates comparing the conventional method by which longer reactive time was consumed. The derivatives were characterized via nuclear magnetic resonance hydrogen spectroscopy (^1^H-NMR), carbon-13 nuclear magnetic resonance spectroscopy (^13^C-NMR), and electrospray ionization high resolution mass spectrometry (ESI-HRMS). The cytotoxicity of these derivatives on *Trichoplusia ni* (Hi-5) cell and *Spodoptera litura* cell (SL cell) was evaluated by 3-(4,5-dimethylthiazol-2-yl)-2,5-diphenyltetrazolium bromide (MTT) bioassays. The results indicated that several compounds had potential cytotoxicity on Hi-5 cell, especially a 4-ethyl substituted alkynyl Schiff base derivative (**3f**) that was demonstrated to possess high selective toxicity to the Hi-5 cell than the SL cell. In addition, **3f** exhibited comparable toxic activity to commercial fipronil on a Hi-5 cell while a little toxic effect on the SL cell, which satisfied the expectation for selective toxicity screening.

## 1. Introduction

Fipronil (**I**) (Figure 1), with two key functional groups of amino and trifluoromethyl sulfonyl group on its arylpyrazole structure, has special insecticidal activity on soil insects and fleas [1,2,3]. Unfortunately, it is difficult to avoid the non-specific toxicity toward aquatic invertebrates and non-target organisms such as honeybees, terrestrial game birds, and shrimp [4,5,6]. In addition, the metabolites derived from fipronil were toxic as well [7,8,9], which would threaten the balance of ecosystems. As a result, fipronil was a restricted application in China and was prohibited from commercial registration in California [10]. Derivation of fipronil to discover compounds with selective toxicity would be an alternative to overcome the dilemma of non-specific toxicity. Hence, it is worth to designate, synthesize, and screen fipronil derivatives with selective toxicity.

It was reported that the 4-trifluoromethylsulfinyl and 5-amine groups at the pyrazole ring of fipronil played an important role in insecticidal activity [2,11,12,13]. Particularly, John E. Casida’s group synthesized a series of fipronil derivatives (**1a**–**1k**) (Figure 1) by varying the 4-position of the pyrazole ring of fipronil and demonstrated that 4-tert-butyl (**1k**), 4-isopropyl (**1h**), and 4-ethyl substituent (**1f**) all exhibited high activities [12,13]. In addition, a series of fipronil derivatives (**II**–**V**) (Figure 1) derived from the active groups were synthesized as a new kind of insecticidal compounds [14,15,16,17]. And among them, vaniliprole (**VI**) (Figure 1) with Schiff base structure derived from 5-amino group was verified to be a kind of new insecticidal derivative, which is less harmful to the environment and user. Compounds containing an alkynyl group usually possess anti-tumor, anti-inflammatory, and anti-oxidation biological activities [18,19]. Series of alkynyl compounds such as propargite [20], oxadiargyl [21], and mepanipyrim [22] were applied as unique insecticides, fungicides, and herbicides, respectively. Hence, introduction of alkynyl Schiff base into the 5-amino group of fipronil would endow it with higher bioactivities. However, insecticides with alkynyl Schiff base in its framework are still relatively rare, especially because the designation of fipronil with 5-alkynyl Schiff base modification has not yet been reported.

On the other hand, selecting an appropriate reaction solvent is essential for organic synthesis. Recently, ionic liquids function as an organic salt, which are composed entirely of ions with melting points below an ambient or a reaction temperature, are attracting numerous attention [23,24]. Compared with conventional organic solvents, ionic liquids possessing tunable solubility, high viscosity, and low vapor pressure functions as “green” solvent in homogeneous synthesis [25,26,27,28]. In addition, ionic liquids can serve not just as a reaction media but as a catalyst as well, which can result in accelerating reaction time and/or improving the yields [24,29,30]. Besides, as green solvents, ionic liquids can also be separated readily from the reaction mixture by simple phase separation and then be regenerated and reused [24]. In this respect, syntheses of compounds in ionic liquids is of great significance to acquire highly synthetic efficiency in line with the concept of environmentally-friendly synthesis.

Encouraged by those descriptions above, in order to screen insecticides with proper selective toxicity and minimal non-target toxicity performance, a series of 4-substituted-*N*-arylpyrazole Schiff bases derivatives **3a**–**3k** (Scheme 1) were synthesized in ionic liquids and conventional solvent. The cytotoxic activity of all designed compounds was evaluated, and a preliminary structure-activity relationship was concluded. The 4-ethyl-*N*-arylpyrazole Schiff base derivative (**3f**) was demonstrated to possess high selective toxicity to the *Trichoplusia Ni* (Hi-5) cell than the *Spodoptera litura* cell (SL cell). The results obtained could render a principle to understand the cytotoxicity for these types of compounds and provide theoretical supports and references for fipronil reconstruction.

## 2. Results and Discussion

### 2.1. Synthesis of 4-Substituted-N-arylpyrazole Schiff Bases ***3a**–**3k***

The new derivatives (**3a**–**3k**) were synthesized by 4-ethynylbenzaldehyde (**2**) and 4-substituted 5-amino-*N*-arylpyrazole (**1a**–**1k**) via a nucleophilic addition elimination reaction, as shown in Scheme 1. In practice, 4-ethynylbenzaldehyde (**2**) was synthesized by cross-coupling reaction [31], and **1a**–**1k** were synthesized as described in the work [13]**.** All structures of the derivatives **3a**–**3k** were characterized via ^1^H-NMR, ^13^C-NMR spectroscopy, and ESI high resolution mass spectrometry.

Attempts to getting efficient and environmentally-friendly synthesis of the derivatives, different reactive solvents, such as ionic liquids (for 48 h) and traditional toluene (for 168 h) was applied to generate **3a**–**3k,** and the synthetic yield of each compound were depicted in Figure 2. Alternatively, the traditional method was conducted in refluxing toluene for seven days with cation exchange resin as an acidic catalyst. As shown in Figure 2a, the synthetic yield tendency obtained in a different solvent was identical. The yield of each compound gained in ionic liquids was beyond 50%, which was much higher than the traditional method by which longer reactive times was consumed. Actually, as shown in Figure 2b, every compound exhibited over 1.5 times of the yield amplification once the experiment was carried out in ionic liquids. It was likely ascribed to the high catalytic efficiency of ionic liquids [26,27,32]. All the desired derivatives could be synthesized by the traditional method, as shown in Figure 2a. It was time-consuming and ineffective when compared to the ionic liquids assisted method. In all, 4-substituted-*N*-arylpyrazole Schiff bases **3a**–**3k** were properly synthesized with more than a 50% yield in ionic liquids. 

### 2.2. Biological Studies

Stably transformed insect cell lines have been applied for high-throughput drug screening and novel insecticides, with receptors as well as ligands discovered in a pest controlling field [33,34,35]. In this study, Hi-5 cell and SL cell were applied to evaluate the cytotoxicity of compounds **3a**–**3k** by MTT assays, which employs fipronil as a positive insecticide. The corresponding inhibition rates of the derivatives were summarized in Table 1. The results indicated that the inhibition efficiency against insect cells differ from each other. **3a**–**3i** display the higher inhibition rate against Hi-5 cell than the SL cell. In general, the compounds exhibit relatively weak activity (<50%, inhibition rate) to the SL cell except for compounds **3j** and **3k**. By increasing the bulk or radius of the halogen atom at 4-position of pyrazole, the inhibition rate against the Hi-5 cell of the compounds **3b**–**3d** increased successively. The 4-ethyl substituted-*N*-arylpyrazole Schiff bases derivative (**3f**) possesses the highest selective toxicity against the Hi-5 cell (96.80%) than the SL cell (5.22%), and exhibited comparable toxic activity to commercial fipronil on the Hi-5 cell. Particularly, the inhibition rates of 4-alkyl derivatives **3e**–**3g** and **3i** for the Hi-5 cell decreased when the straight-chain alkyl substitute was greater over two carbon atoms, which indicates that extending the alkyl chain length was not beneficial to acquire highly toxic activity. However, according to **3g**, **3h**, and **3i**–**3k**, the branched chain carbon atoms of the 4-alkyl substitute enlarged, the inhibition rates of the compounds against Hi-5 cell decreased, and toxic activity against the SL cell increased. The above results indicated that the electron density of halogen and the hindrance of bulky alkyl for 4-substituted-*N*-arylpyrazole Schiff bases played an important role in selective cellular toxicity. In all, a series of 4-substituted-*N*-arylpyrazole Schiff bases derivatives were synthesized and was proven to be able to seek selective cellular toxicity.

## 3. Materials and Methods 

### 3.1. Synthesis

#### 3.1.1. General Information

4-Butyl-3-methylimidazolium tetrafluoroborate (ionic liquids) was obtained from Shanghai Cheng Jie Chemical Co., Ltd. (China). 4-bromobenzaldehyde was purchased from Aladdin Reagent Co., Ltd. (China). Trimethylsilylacetylene (>98%) was obtained from Shanghai Rui Yi Medical Tech Co., Ltd. (China). Palladium (II) acetate (Pd(OAc)_2_), cuprous (Ι) iodide (CuI), triphenylphosphine (PPh_3_), and 3-(4,5-dimethylthiazol-2-yl)-2,5-diphenyltetrazolium bromide (MTT) were acquired from Sigma-Aldrich Chemical Co., (China). Column chromatography (CC): silica gel (100–200, or 200–300 mesh, Qingdao Marine Chemical Factory) was used for purification. The melting points (M.p.) of the newly synthesized compounds were tested on an WRR digital melting point apparatus. The ^1^H-NMR and ^13^C-NMR spectra were recorded on a Bruker AV-600 spectrometer, using tetramethylsilane (TMS) as an internal standard, δ in ppm, J in Hz. ESI-HRMS was collected with JEOL JMS-T100CS cold-spray time-of-flight mass spectrometer, in *m*/*z*. The enzyme-linked immunosorbent assay (ELISA) microplate-reader (Bio-Rad) was applied for cytotoxicity experiments.

#### 3.1.2. Synthesis of 4-ethynylbenzaldehyde (**2**)

The 4-ethynylbenzaldehyde (**2**) was synthesized by a cross-coupling reaction [31] and the corresponding conventional preparation process was shown in Scheme 1. Pd(OAc)_2_, CuI, and PPh_3_ (1:2:3 equiv) was added into dried Et_3_N under nitrogen and stirred for 10 min. Subsequently, 4-bromobenzaldehyde (**A**) and trimethylsilylacetylene (**B**) (1.5:1 equiv) were added into the former reaction mixture, which would be turned to red and faded away quickly. Then, the above solution was heated to 80 °C and stirred for 2–3 h. After that, the precipitate was filtrated, washed with 1 M hydrochloric acid and water, and dried with anhydrous MgSO_4_. The solvent was removed in vacuo and the residue was separated by CC (SiO_2_, petroleum ether/EtOAc 10:1) to acquire **C** as a yellow crystal. Lastly, it was deprotected as reported previously to give 4-ethynylbenzaldehyde (**2**) [31]. Yellow crystals. Yield 84%. M.p.69–70 °C. ^1^H-NMR (600 MHz, CDCl_3_) *δ* 3.30 (*s*, 1H, CHO), 7.63 (*d*, 2H, *J* = 8.4 Hz*,* H-Ph), 7.83 (*d*, 2H, *J* = 8.4 Hz, H-Ph), 10.01 (*s*, 1H, NH); ^13^C-NMR (150 Hz, Acetone-*d*_6_) *δ* 83.0, 83.5, 129.1, 130.5, 133.6, 137.5, 192.4. 

#### 3.1.3. Synthesis of 4-Substituted *N*-arylpyrazole Schiff Base (**3a**–**3k**)

General procedure for synthesis of 4-substituted *N*-arylpyrazole Schiff base (**3a**–**3k**) was described in Scheme 1. The 4-substituted 5-amino*-N-*arylpyrazole compounds **1a**–**1k** were synthesized, according to previous literature [13]. The specific operation procedures for **3a**–**3k** were as follows.
Ionic liquids assisted synthesis. The 4-substituted-5-amino*-N-*arylpyrazole compounds **1a**–**1k** (0.46 mmol) were first dissolved into the ionic liquids (3 mL), then 4-ethynylbenzaldehyde (**2**) (0.38 mmol) and hydrochloric acid (6 M) was added into the above mixture successively and, subsequently, stirred for 48 h. After the reaction finished, the reaction mixture was extracted with Et_2_O, the upper phase was concentrated, and purified by CC to get the target compounds. In addition, the lower phase was the ionic liquids because of its low solubility in Et_2_O, and it was concentrated by vacuum distillation for recycling use.Conventional solvent synthesis. The 4-substituted-5-amino*-N-*arylpyrazole compounds **1a**–**1k** (0.46 mmol) and 4-ethynylbenzaldehyde (**2**) (0.38 mmol) were dissolved in toluene, following 1 g Resin 732 and 1 g 4 Å molecular sieve were added. The reaction mixture was refluxed 168 h, and then cooled to room temperature and filtered. The filtrate was evaporated in vacuo. The residue was purified by silica gel column chromatography (petroleum ether/EtOAc) to get the target product.

The yield and the properties of the products as well as the analytical data of nuclear magnetic resonance hydrogen spectroscopy (^1^H-NMR), carbon-13 nuclear magnetic resonance spectroscopy (^13^C-NMR), and electrospray ionization high resolution mass spectrometry (ESI-HRMS) were listed as follows. The ESI-HRMS spectra of the synthesized derivatives were appended in Appendix A.

*1-(2,6-dichloro-4-(trifluoromethyl)phenyl)-5-(4-ethynylbenzylideneamino)-1H-pyrazole-3-carbonitrile* (**3a**). Yellow powder. Yield 88%. M.p. 189–191 °C. ^1^H-NMR (600 MHz, Acetone-d6) δ 3.91(s, 1H, CH≡), 7.31 (s, 1H, H-pyrazole), 7.58 (d, 2H, *J* = 7.2 Hz, H-Ph), 7.82 (d, 2H, *J* = 7.8 Hz, H-Ph), 8.13 (s, 2H, H-Ph), 9.09(s, 1H, CH=N), ^13^C-NMR (150 MHz, Acetone-d6) δ 82.6, 83.5, 99.1, 114.2, 122.5 (q, *J* = 271 Hz), 127.0, 127.7, 128.3, 130.3, 133.3, 134.4 (q, *J* = 34 Hz), 136.0, 136.6, 138.0, 153.5, 165.9; ESI-HRMS calcd for C_20_H_9_Cl_2_F_3_N_4_ [M + H]^+^ 433.0234, found, 432.9991 (Appendix A).

*4-chloro-1-(2,6-dichloro-4-(trifluoromethyl)phenyl)-5-(4-ethynylbenzylidene-amino)-1H-pyrazole -3-carbonitrile* (**3b**). Yellow powder, Yield 59%, M.p. 164–166 °C. ^1^H-NMR (600 MHz, Acetone-d6) δ 3.96 (s, 1H, CH≡), 7.60 (d, 2H, *J* = 8.4 Hz, H-Ph), 7.88 (d, 2H, *J* = 8.4 Hz, H-Ph), 8.15 (s, 2H, H-Ph), 9.41 (s, 1H, CH=N), ^13^C-NMR (150 MHz, Acetone-d6) δ 83.0, 83.4, 104.0, 111.8, 122.4 (q, *J* = 272 Hz), 127.2, 128.0, 128.3, 130.5, 133.4, 134.8 (q, *J* = 34 Hz), 135.9, 136.4, 137.6, 146.9, 167.5, ESI-HRMS calcd for C_20_H_8_Cl_3_F_3_N_4_ [M + H]^+^, 466.9845, found, 466.9529 (Appendix A).

*4-bromo-1-(2,6-dichloro-4-(trifluoromethyl)phenyl)-5-(4-ethynylbenzylidene-amino)-1H-pyrazole -3-carbonitrile* (**3c**). Yellow powder, Yield 82%, M.p. 187–189 °C. ^1^H-NMR (600 MHz, Acetone-d6) δ 3.95 (s, 1H, CH≡), 7.60 (d, 2H, *J* = 8.4 Hz, H-Ph), 7.88 (d, 2H, *J* = 8.4 Hz, H-Ph), 8.14 (s, 2H, H-Ph), 9.41 (s, 1H, CH=N), ^13^C-NMR (150 MHz, Acetone-d6) δ 83.0, 83.4, 87.9, 112.5, 122.4 (q, *J* = 271 Hz), 127.2, 128.3, 130.2, 130.5, 133.4, 134.7 (q, *J* = 32 Hz), 135.7, 136.4, 137.6, 148.5, 167.7, ESI-HRMS calcd for C_20_H_8_BrCl_2_F_3_N_4_ [M + H]^+^, 510.9348, found, 510.8907 (Appendix A).

*4-iodo-1-(2,6-dichloro-4-(trifluoromethyl)phenyl)-5-(4-ethynylbenzylidene-amino)-1H-pyrazole -3-carbonitrile* (**3d**). Yellow powder, Yield 59%, M.p. 186–188 °C. ^1^H-NMR (600 MHz, Acetone-d6) δ 3.95 (s, 1H, CH≡), 7.61 (d, 2H, *J* = 8.4 Hz, H-Ph), 7.89 (d, 2H, *J* = 8.4 Hz, H-Ph), 8.12 (d, 2H, H-Ph), 9.38 (s, 1H, CH=N), ^13^C-NMR (150 MHz, Acetone-d6) δ 53.3, 83.0, 83.4, 113.7, 122.4 (q, *J* = 271 Hz), 127.2, 128.3, 130.5, 130.6, 133.4, 134.5 (q, *J* = 34 Hz), 135.5, 136.3, 137.7, 151.4, 168.0, ESI-HRMS calcd for C_20_H_8_Cl_2_F_3_IN_4_ [M + H]^+^, 558.9210, found, 558.8715 (Appendix A).

*1-(2,6-dichloro-4-(trifluoromethyl)phenyl)-5-(4-ethynylbenzylideneamino)-4-methyl-1H-pyrazole -3-carbonitrile* (**3e**). Yellow powder, Yield 60%, M.p. 163–164 °C. ^1^H-NMR (600 MHz, Acetone-d6) δ 2.50 (s, 3H, CH_3_), 3.91 (s, 1H, CH≡), 7.56 (d, 2H, *J* = 8.4 Hz, H-Ph), 7.82 (d, 2H, *J* = 7.8 Hz, H-Ph), 8.10 (s, 2H, H-Ph), 9.05 (s, 1H, CH=N), ^13^C-NMR (150 MHz, Acetone-d6) δ 9.4, 82.5, 83.5, 111.2, 113.7, 122.4 (q, *J* = 270 Hz), 127.0, 127.5, 129.6, 130.0, 133.2, 134.2 (q, *J* = 34 Hz), 136.5, 138.4, 148.5, 165.1, ESI-HRMS calcd for, C_21_H_11_Cl_2_F_3_N_4_ [M + H]^+^, 447.0400, found, 447.0060 (Appendix A).

*1-(2,6-dichloro-4-(trifluoromethyl)phenyl)-5-(4-ethynylbenzylideneamino)-4-ethyl-1H-pyrazole -3-carbonitrile* (**3f**). Yellow powder, Yield 56%, M.p. 167–168 °C. ^1^H-NMR (600 MHz, Acetone-d6) δ1.36 (t, 3H, *J* = 7.2 Hz, CH_3_), 2.89 (q, 2H, *J* = 7.2 Hz, CH_2_CH_3_), 3.91 (s, 1H, CH≡), 7.57 (d, 2H, *J* = 7.8 Hz, H-Ph), 7.84 (d, 2H, J = 8.4 Hz, H-Ph), 8.09 (s, 2H, H-Ph), 8.94 (s, 1H, CH=N); ^13^C-NMR (150 MHz, Acetone-d6) δ 14.9, 17.4, 82.5, 83.5, 113.9, 117.3, 122.4 (q, *J* = 271 Hz), 127.0, 127.6, 128.7, 130.1, 133.2, 134.2 (q, *J* = 34 Hz), 136.3, 136.5, 138.4, 148.6, 165.5; ESI-HRMS calcd for C_22_H_13_Cl_2_F_3_N_4_ [M + H]^+^, 461.0547, found, 461.0213 (Appendix A).

*1-(2,6-dichloro-4-(trifluoromethyl)phenyl)-5-(4-ethynylbenzylideneamino)-4-propyl-1H-pyrazole-3-carbonitrile* (**3g**). Yellow powder, Yield 63%, M.p. 128–130 °C. ^1^H-NMR (600 MHz, Acetone-d6) δ 1.02 (t, 3H, *J* = 7.2 Hz, CH3), 1.79 (sext, *J* = 7.2 Hz, CH_2_CH_2_CH_3_), 2.87 (t, 2H, *J* = 7.2 Hz, CH_2_CH_2_CH_3_), 3.96 (s, 1H, CH≡), 7.57 (d, 2H, *J* = 7.8 Hz, H-Ph), 7.83 (d, 2H, *J* = 8.4 Hz, H-Ph), 8.10 (s, 2H, H-Ph), 8.96 (s, 1H, CH=N), ^13^C-NMR (150 MHz, Acetone-d6) δ 13.8, 23.8, 25.7, 82.5, 83.5, 114.0, 115.7, 122.4 (q, *J* = 271 Hz), 127.0, 127.6, 129.2, 130.1, 133.3, 134.2 (q, *J* = 34 Hz), 136.3, 136.5, 138.4, 148.8, 165.4, ESI-HRMS calcd for C_23_H_15_Cl_2_F_3_N_4_ [M + H]^+^, 475.0740, found, 475.0258 (Appendix A).

*1-(2,6-dichloro-4-(trifluoromethyl)phenyl)-5-(4-ethynylbenzylideneamino)-4-isopropyl-1H-pyrazole -3-carbonitrile* (**3h**). Yellow powder, Yield 53%, M.p. 158–160 °C. ^1^H-NMR (600 MHz, Acetone-d6) δ 1.47 (s, 3H, CH_3_), 1.48 (s, 3H, CH_3_), 3.24 (sept, 1H, *J* = 7.2 Hz, CH(CH_3_)2), 3.92 (s, 1H, CH≡), 7.57 (d, 2H, *J* = 8.4 Hz, H-Ph), 7.84 (d, 2H, *J* =7.8 Hz, H-Ph), 8.07 (s, 2H, H-Ph), 8.80 (s, 1H, CH=N), ^13^C-NMR (150 MHz, Acetone-d6) δ 23.3, 24.7, 82.6, 83.5, 114.8, 121.6, 124.2 (q, *J* = 271 Hz), 127.0, 127.7, 129.4, 130.2, 133.2, 134.4 (q, *J* = 35 Hz), 136.5, 138.2, 147.3, 148.5, 166.8, ESI-HRMS calcd for C_23_H_15_Cl_2_F_3_N_4_ [M + H]^+^, 475.0740, found, 475.0382 (Appendix A).

*1-(2,6-dichloro-4-(trifluoromethyl)phenyl)-5-(4-ethynylbenzylideneamino)-4-butyl-1H-pyrazole -3-carbonitrile* (**3i**). Yellow powder, Yield 82%, M.p. 106–108 °C. ^1^H-NMR (600 MHz, Acetone-d6): δ 0.96 (t, 3H, *J* = 7.2 Hz, CH_3_), 1.46 (sext, 2H, *J* = 7.8 Hz, CH_2_CH_2_CH_2_CH_3_), 1.74 (quint, 2H, *J* = 7.8 Hz, CH_2_CH_2_CH_2_CH_3_), 2.88 (t, 2H, *J* = 7.8 Hz, CH_2_CH_2_CH_2_CH_3_), 3.91 (s, 1H, CH≡), 7.56 (d, 2H, *J* = 8.4 Hz, H-Ph), 7.82 (d, 2H, *J* = 8.4 Hz, H-Ph), 8.09 (s, 2H, H-Ph), 8.95 (s, 1H, CH=N), ^13^C-NMR (150 MHz, Acetone-d6) δ 14.0, 22.9, 23.6, 32.6, 82.5, 83.5, 114.0, 116.0, 122.4 (q, *J* = 271 Hz), 127.0, 127.6, 129.1, 130.1, 133.3, 134.2 (q, *J* = 34 Hz), 136.3, 136.5, 138.4, 148.7, 165.4, ESI-HRMS calcd for C_24_H_17_Cl_2_F_3_N_4_ [M + H]^+^, 489.0860, found, 489.3054 (Appendix A).

*1-(2,6-dichloro-4-(trifluoromethyl)phenyl)-5-(4-ethynylbenzylideneamino)-4-isobutyl-1H-pyrazole -3-carbonitrile* (**3j**).Yellow powder, Yield 58%, M.p. 94–95 °C. ^1^H-NMR (600 MHz, Acetone-d6) δ 1.00 (s, 3H, CH_3_), 1.01 (s, 3H, CH_3_), 2.27-2.30 (m, 1H, CH_2_CH(CH_3_)2), 2.78 (d, 2H, *J* = 7.2 Hz, CH_2_CH(CH_3_)2), 3.91 (s, 1H, CH≡), 7.56 (d, 2H, *J* = 8.4 Hz, H-Ph), 7.81 (d, 2H, *J* = 8.4 Hz, H-Ph), 8.10 (s, 2H, H-Ph), 8.98 (s, 1H, CH=N), ^13^C-NMR (150 MHz, Acetone-d6) δ 22.5, 23.3, 30.4, 32.6, 82.6, 83.5, 114.1, 115.0, 122.4 (q, *J* = 271 Hz), 127.0, 127.6, 129.6, 130.15, 133.3, 134.4 (q, *J* = 34 Hz), 136.3,136.5, 138.4, 148.9, 165.4, ESI-HRMS calcd for C_24_H_17_Cl_2_F_3_N_4_ [M + H]^+^, 489.0860, found, 489.0541 (Appendix A).

*1-(2,6-dichloro-4-(trifluoromethyl)phenyl)-5-(4-ethynylbenzylideneamino)-4-tert-butyl-1H-pyrazole-3-carbonitrile* (**3k**). Yellow powder, Yield 59%, M.p. 125–127 °C. ^1^H-NMR (600 MHz, Acetone-d6) δ 1.49 (s, 9H, 3CH_3_), 3.92 (s, 1H, CH≡), 7.58 (d, 2H, *J* = 8.4 Hz, H-Ph), 7.82 (d, 2H, *J* = 8.4 Hz, H-Ph), 8.04 (s, 2H, H-Ph), 8.45 (s, 1H, CH=N), ^13^C-NMR (150 MHz, Acetone-d6) δ 31.3, 32.0, 82.7, 83.4, 115.6, 122.3 (q, *J* = 271 Hz), 124.5, 126.8, 127.3, 128.0, 130.2, 133.4, 134.4 (q, *J* = 34 Hz), 135.7, 136.6, 137.8, 148.8, 167.8. ESI-HRMS calcd for C_24_H_17_Cl_2_F_3_N_4_ [M + H]^+^, 489.0860, found, 489.2940 (Appendix A).

### 3.2. Biological Studies

#### 3.2.1. Cell Culture

High 5 Cell lines (Hi-5) derived from embryos of *T. ni*, and SL cell lines derived from the ovary of *S. litura*, were obtained from the Key Laboratory of Pesticide and Chemical Biology of Ministry of Education, Central China Normal University, China. Two cell lines were cultured in Grace’s insect cell culture medium (Gibco, America) and supplemented with 10% fetal calf serum at 27.5 °C. Cells in the logarithmic phase of growth were used in the experiments. All the tested compounds were dissolved in dimethyl sulfoxide (DMSO) and then diluted to the desired concentrations with cell culture medium before use. The concentration of DMSO was kept 0.1% in treated groups. Control cultures were performed in the presence of 0.1% DMSO under the same culture conditions [34].

#### 3.2.2. Cytotoxicity of the Compounds in Cell Lines

The cytotoxicity of compounds **3a**–**3k** were measured with MTT assays referring to previously literature [33,34]. The cells were first seeded in 96-well plates for one day, and then incubated with 20 μg·mL^−1^ of the compounds for 48 h, among which DMSO was used as a negative control and fipronil was used as a positive control. After that, the cultural medium was replaced with MTT (0.5 mg·mL^−1^) solution and the cells were incubated for 4 h at 27.5 °C. Lastly, the produced formazan precipitate was dissolved with 200 μL DMSO and the absorbance values at 570 nm were recorded on a Bio-Rad ELISA microplate-reader. The inhibition rate was calculated by the absorbance.

#### 3.2.3. Statistical Analysis

All the experiments were repeated at least three times. Data were expressed as the means ± SE and analyzed by Duncan’s One-Way Analysis of Variance (ANOVA). The level of significance was set at *p* < 0.05.

## 4. Conclusions

In conclusion, a series of novel Schiff bases **3a**–**3k** with an alkynyl group was designed and synthesized by 4-ethynylbenzaldehyde (**2**) and 4-substitued 5-amino-*N*-arylpyrazole (**1a**–**1k**) in ionic liquids with more than a 50% yield. Results from MTT bioassays on Hi-5 cell and SL cell indicated that the synthesized compounds showed selective cellular toxicity. Particularly, the 4-ethyl-*N*-arylpyrazole Schiff base derivative (**3f**) was demonstrated to possess high selective toxicity with comparable toxic activity to commercial fipronil on the Hi-5 cell while a little toxic effect on the SL cell. In all, the alkynyl Schiff base derivation and 4-ethyl substitution for fipronil satisfy the expectation for selective toxicity insecticide screening, which would provide theoretical supports and references for fipronil reconstruction. Nonetheless, for a practical application, the insecticidal activities in vivo of the derivatives are expected and the mechanism of the selective toxicity is worth further investigation.

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
