# Peer review of "Ionic Liquids Enhanced Alkynyl Schiff Bases Derivatives of Fipronil Synthesis and Their Cytotoxicity Studies"

_molecules, 2019, doi:10.3390/molecules24183223_

Round 1

Reviewer 1 Report

Xiu Liu, Linya Huang, Hongjun Chen, Na Li, Chao Yan, Chenzhong Jin, Hanhong Xu

Ionic liquid enhanced Alkynyl Schiff bases 2 derivatives of fipronil synthesis and their cytotoxicity 3 studies

Ms 590049

1. From the chemical (synthetical) point of view, the proposed manuscript brings little novelty. The proposed work is limited to the synthesis of Schiff bases 3a-3k in two different solvents (toluene, IL) of fipronil derivatives 2a-2k which synthesis has been described in [35]. The work [35] is referred to in the manuscript as "previous literature" (line 139). This suggests that this is the previous work of the manuscript 590049 authors. This is not the case. This is the work of another research group (USA). The authors of the manuscript repeated the results of the synthesis of the described in [35] compounds, and then modified them with the aldehyde 1. Work [35] is cited, so everything is OK. Nevertheless, the proposed manuscript looks strange to some extent, when it becomes clear that the main experimental work is a repetition of [35], and the authors' synthetic effort is focused only on the synthesis of the Schiff bases 3a-3k. In conclusion, it seems to me that the mandatory condition for accepting work in "Molecules" is a significant rebuilding of the manuscript:

- the work [35] must be cited in the introduction; moreover it must be clearly stated, that the aldehyde 1 is used to the modification of 2a-2k which synthesis has been described in the work [35].

- the procedure 3.1.2 (lines 137-146) and scheme 1 should be removed. Scheme 1 is almost identical to the scheme presented in [35]. In the experimental part the synthesis of 2a-2k compounds should be referred to as “synthesized according to [35]”.

- Scheme 2 (3a-3k synthesis) belongs to the "Results" section and should be placed there.

2. ESI-MS is defined as „m/z (%)”. I can not recognize % data.

3. There are neither elemental analyses nor HR MS data for the synthesized compounds.

4 The separation of ionic liquids from reaction mixtures or products can be difficult. It would be advisable to insert a comment devoted to this problem.

5.References: pages should be indicated as 3345-3355, not 3345-55.

6. tobluene --> toluene

Author Response

We would like to thank you for giving us constructive suggestions which would help us to improve the quality of the paper. The comments were responded point-by-point (please see the attachment). Efforts were also made to correct the mistakes and improve the English of the manuscript.

Reviewer 2 Report

In this work the ionic liquid enhanced alkynyl schiff bases derivatives of fipronil synthesis and their cytotoxicity studies are described. Utilization of ionic liquids demonstrated to endow the yield of each compound beyond 50%, enhance over 1.5 times of the synthetic production comparing with the conventional method where longer reactive time was consumed. The cytotoxicity of these derivatives on Trichoplusia ni (Hi-5) cell and Spodoptera litura cell (SL cell) was evaluated by MTT bioassays. It was found that several compounds had potential cytotoxicity on Hi-5 cell, especially 4-ethyl substituted alkynyl Schiff base derivative was demonstrated to possess high selective toxicity to the Hi-5 cell than SL cell. The article looks like a short communication and may be published after minor revision.

Notes:

In my opinion the schemes of synthesis of target products should be presented in the Results and Discussion section. There are some typographical errors in the article, like “Actone-d6” (line 181), “Aectone-d6” (line 195, 202, 209, 216, 223, 231, 238, 247, 255), “4-substitued” (line 284). They should be checked and corrected. Materials and Methods section. In the description of mass spectra of new compounds the calculated values m/z should be added. Promising application of new obtained results should be added in Conclusion of the work.

Author Response

(The authors gave the same response as above.)

Round 2

Reviewer 1 Report

The manuscript has been corrected sufficiently to publish the paper.

Additional remarks.

ESI HR-MS have been added into the spectral characterization. The presented data are referred to Figures S... without the explanation that they are presented in the Supplementary material. Please add into the text an information about the existence of the Supplementary Material and its content.

"tobluene" still exists in Scheme. Please correct.